# Sex Differences in the Correlation between Fatigue Perception and Regional Gray Matter Volume in Healthy Adults: A Large-Scale Study

**DOI:** 10.3390/jcm11206037

**Published:** 2022-10-13

**Authors:** Handityo Aulia Putra, Kaechang Park, Fumio Yamashita

**Affiliations:** 1Research Organization for Regional Alliances, Kochi University of Technology, Kochi 782-8502, Japan; 2Division of Ultrahigh Field MRI, Institute for Biomedical Sciences, Iwate Medical University, Iwate 028-3694, Japan

**Keywords:** regional gray matter volume, fatigue perception, Chalder’s fatigue questionnaire, myalgic encephalomyelitis/chronic fatigue syndrome, coronavirus disease 2019

## Abstract

The relationship between fatigue perception and regional gray matter volume (rGMV) has seldom been studied in healthy adults. Therefore, this study aimed to analyze sex differences in the correlation between rGMV and fatigue perception using Chalder’s fatigue questionnaire (CFQ). The CFQ was used to analyze the sexual features of rGMV related to the degree of perceived fatigue in 2955 healthy adults (male = 1560, female = 1395) of various ages (20–89 years, median 56). A higher CFQ score denotes a higher perceived fatigue level by the participant. According to the CFQ scores in males, the volumes of the right orbital part of the inferior frontal gyrus and left precuneus were negatively correlated (i.e., smaller rGMV had a higher CFQ score), whereas the left angular gyrus was positively correlated. In females, the right inferior temporal gyrus was negative, whereas the left middle temporal gyrus and right putamen were positive (i.e., larger rGMV had a higher CFQ score). The lack of identified regions in this large-scale study between males and females might be related to sex differences in clinical or pathological fatigue morbidities. Additionally, the sex differences in the negative or positive correlations between rGMV and fatigue perception may contribute to a better understanding of the neuronal mechanism in the early stages of fatigue development.

## 1. Introduction

Fatigue is a common symptom of fever and pain [1]. In recent years, myalgic encephalomyelitis/chronic fatigue syndrome (ME/CFS) has been defined as a persistent and pathological state of fatigue that frequently interferes with daily activities for more than 6 months, including cognitive, immune, and autonomic dysfunctions [2,3]. In some developed countries, such as Japan, “Karoshi”—death from overworking with severe fatigue—is an urgent national health and hygiene issue [4]. Structural studies using magnetic resonance imaging (MRI) in patients with ME/CFS revealed bilateral frontal lobe atrophy and volumetric changes in the hippocampal region, amygdala, insula [5], and basal ganglia [6,7]. During attention-demanding tasks, functional MRI (fMRI) studies with normal healthy individuals revealed decreased signal values in the ventrolateral prefrontal cortex [8] and posterior parietal cortex but increased values in the cerebellar, temporal, cingulate, frontal [9], and medial orbitofrontal cortices [10].

Additionally, fMRI revealed abnormal resting-state functional connectivity between the left anterior mid-cingulate and sensorimotor [11]. Furthermore, a positron emission tomography (PET) study revealed a decrease in regional cerebral blood flow in patients with ME/CFS in various brain regions: frontal, prefrontal, orbitofrontal, middle temporal, superior temporal, transverse temporal, middle occipital, putamen, globus pallidus, and hippocampus [12]. Nevertheless, due to the relatively few patients with ME/CFS in previous studies, there has been a lack of statistical power to investigate the specific brain regions with rather moderate changes or smaller volumes. In addition, early preclinical stage detection and countermeasures are essential to suppress fatigue development. Therefore, examining many healthy adults without ME/CFS is necessary to elucidate the early mechanisms of fatigue progression statistically. In a more recent study, we investigated the correlation between cerebral gray matter volume and fatigue in different work types for 1618 healthy middle-aged participants, which is a different set of participants than the present study [13]. The previous study showed that different work types impacted different regions of the gray matter; however, it targeted middle-aged subjects, and did not further investigate the effect of different sex on the correlation between gray matter volume and fatigue. 

Post-virus fatigue has recently attracted significant attention globally as a sequela of coronavirus disease 2019 (COVID-19). Fatigue is likely to persist for some time after the infection has cleared. It frequently coexists with neurological comorbidities to cause patients to sleep, feel unsteady on their feet, and have difficulty concentrating and memorizing. Sex differences have been recognized as predictors of COVID-19 progression and health outcomes [14]. Male patients have higher COVID-19-related mortality, comorbidity severity, and intensive care unit admissions than female patients [15], whereas the opposite tendency was observed in long COVID-19 syndromes, such as persistent fatigue and somnolence tendency [16]. However, numerous studies have discovered that females have a substantially higher rate of ME/CFS than males, with estimates ranging from 75 to 85% [17]. The morbidity of chronic fatigue caused by COVID-19 or ME/CFS is predominant in females. Thus, sex differences have attracted considerable interest in fatigue progression. Therefore, this study aimed to analyze sex differences in the correlation between regional gray matter volume (rGMV) and fatigue degree using Chalder’s fatigue questionnaire (CFQ) scores on 2955 healthy adults without ME/CFS and COVID-19 infection. 

## 2. Materials and Methods

### 2.1. Participants 

As part of the brain healthcare checkups in the Kochi Kenshin Clinic, affiliated with the Kochi University of Technology, 4140 individuals aged 20–89 years (average age; 53.15 ± 9.74) underwent MRI and completed the questionnaire on fatigue. Each participant gave written informed consent to participate in the project. This study was conducted following the Declaration of Helsinki and approved by the ethics committee of Kochi University of Technology (No. 145).

Figure 1 depicts the conditions under which participants were enrolled. Before the COVID-19 pandemic, 4233 participants were recruited from 2016 to 2018 and examined for any diseases that might affect their brain scans. Twenty-five participants who failed the brain disease health screening were excluded from the study. The remaining 4208 participants were asked to answer the translated version of the CFQ following the work of Tanaka et al. [17]. Sixty-eight participants were excluded from the study because they did not correctly answer all CFQ questions. Finally, 4140 participants were processed for the MRI scans. The participants were instructed to stay still in a supine position during the MRI scan; however, all images were blurry because of noise movement. Space-occupying lesions were identified, such as brain tumors and arachnoid cysts, which interfere with volumetric measurements. We excluded participants with small-vessel disease, including small subcortical (lacunar) infarcts (of deep gray nuclei and deep white matter), hemorrhages, perivascular spaces, and diffuse white matter changes. We also excluded participants with hypothyroidism and hyperthyroidism and taking neurotropic drugs such as antidepressants. Thus, the MRI scans of 1185 participants were excluded from the study after a series of image quality checks for volumetric analysis. Finally, 2955 participants (53.15 ± 9.74 years) were registered for statistical analysis (Figure 1).

### 2.2. Assessment of Fatigue

The degree of fatigue was evaluated using the Japanese-translated version of the CFQ [18], a self-administered questionnaire used to measure the extent and severity of fatigue in both clinical and non-clinical populations. The questionnaire was initially developed to measure the severity of chronic fatigue syndrome (CFS) in the clinical population [19]; however, the scale has been revised and is now more widely used to measure tiredness in the non-clinical population [13,20,21]. 

The questions are benign, non-threatening, and concerned with sensation and functionality. Each of the 11 items is graded on a 4-point scale from asymptomatic to maximum symptomologies, such as “Better than usual”, “No worse than usual”, “Worse than usual”, and “Much worse than usual”. For all items, the least symptomatic answers were on the left side of the response set, providing respondents with an easy-to-understand checklist. The global score ranges from 0 to 33, with 0 indicating no fatigue and 33 showing extreme fatigue.

The reliability coefficients for the CFQ have been discovered to be high in studies including patients with CFS [22] and occupational and general population research. The CFQ has been widely used in studies on fatigue involving the working population, and it consistently outperforms other longer and multidimensional tools [23]. Additionally, the CFQ has been widely used in occupational research, and it allows simple comparisons between studies and populations, another advantage of using this tool.

### 2.3. MRI

T_1_-weighted MRI images were obtained using a 1.5 Tesla ECHELON Vega system (Hitachi, Tokyo, Japan) with a three-dimensional gradient-echo inversion recovery sequence. The following scanning parameters were used: echo time, 4.0 ms; repetition time, 9.2 ms; inversion time, 1000 ms; flip angle, 8°; field of view, 240 mm; matrix size, 0.9375 × 0.9375 mm; slice thickness, 1.2 mm; and the number of excitations, 1. Each image was visually assessed for brain disease, anomalies, head motion, and artifacts affecting volumetric measurements. The images were processed and analyzed to estimate the rGMV using the VBM8 toolbox (http://dbm.neuro.uni-jena.de/vbm8/, accessed on 1 August 2022) and other modules implemented in the statistical parametric mapping (SPM) 8 (https://www.fil.ion.ucl.ac.uk/spm/, accessed on 1 August 2022).

Briefly, the images were segmented into gray matter (GM), white matter (WM), and cerebrospinal fluid (CSF) spaces using the maximum a posteriori approach [24,25]. Using a high-dimensional nonlinear warping algorithm, the segmented GM and WM images were then used to estimate the morphological correspondence between the template image and the participant’s brain [26]. The estimated non-linear warp was inversely applied to an atlas defined in the template space to parcellate the target brain anatomically. The SPM12 neuromorphometrics atlas was used for parcellation, with a modification for WM lesions, which appeared as incorrect GM segments around the lateral ventricles. The volume of each anatomical region was the sum of the corresponding tissue densities in the voxels belonging to that region.

### 2.4. Statistical Analysis

The correlation between fatigue and brain volume (overall GM and CSF volumes and total intracranial volume (ICV), including regional volumes for the 110 brain regions), as estimated using the SPM8 toolbox, was investigated using bivariate correlation in Statistical Package for the Social Sciences (SPSS) version 22 (IBM Corp., Armonk, NY, USA). 

Furthermore, an exploratory analysis of the relationship between segment volume or rGMV and fatigue was performed using multiple linear regression with a confidence interval of 95%. Multiple linear regression analyses were performed separately and independently for each group. The three groups represented the entire group as well as males and females separately. The dependent variable was the total fatigue score, while age and rGMV were the independent variables because age affects brain volume [26]. The volume data used are rGMVs that have already been divided by the ICV of the brain; this was done to minimize individual variation in brain size. 

## 3. Results

### 3.1. Age, CFQ Scores, and Ratios of Several Brain Volumes to ICVs

Figure 2 shows that the distribution of the CFQ score follows a normal distribution for both sex groups. Additionally, Table 1 presents the participants’ mean and standard deviation (SD) for age, CFQ, and brain regions, including total brain volume, total gray matter volume (GMV), and total white matter volume. The CFQ score was highly reliable (14 items; Cronbach’s α = 0.89). The female CFQ scores were significantly higher than the male scores (Table 1). Total brain volume/ICV, total GMV/ICV, and total WMV/ICV were also significantly higher in females than males (Table 1).

### 3.2. Analyses of rGMV

Bivariate correlation using Pearson’s coefficient was performed for total fatigue scores, sex, age, and 110 rGMVs. The correlation with the Bonferroni correction (confidence interval = 95%, Bonferroni-adjusted *p* = 0.05/113 = 0.0004 < 0.001) showed that the total fatigue score significantly correlates with sex (Pearson = −0.152, *p* < 0.001) and total fatigue score with age (Pearson = −0.133, *p* < 0.001). Additionally, approximately half of the 110 rGMVs, including the rGMVs presented in Table 2, are significantly correlated with the total fatigue score, except for the right orbital part of the inferior frontal gyrus (Pearson = 0.017, *p* = 0.344). 

Multiple linear regression analysis was performed for total fatigue based on sex, age, and 110 rGMVs. For three linear regressions, linearity was assessed using partial regression plots and a plot of studentized residuals against the predicted values. Durbin–Watson statistics of 1.999, 2.074, and 1.960 for all participants, male participants, and female participants, respectively, demonstrated that residuals were independent. Homoscedasticity was assessed using visual inspection of a plot of studentized residuals versus unstandardized predicted values. Tolerance values > 0.1 revealed no evidence of multicollinearity. There were no studentized deleted residuals greater than ±3 SDs, no leverage values greater than 0.2, and Cook’s distance values above 1. A Q–Q plot demonstrated that the assumption of normality was met. A multiple regression analysis was run to predict total fatigue in all participants from sex, age, and 110 rGMVs. These variables statistically significantly predicted total fatigue (F(4, 2950) = 33.964, *p* < 0.0001). Similar analyses were done for male participants and female participants. The inputted variables statistically significantly predicted the total fatigue in male participants (F(4, 1555) = 9.938, *p* < 0.0001) and female participants (F(3, 1391) = 15.691, *p* < 0.0001). Table 2 presents the regression coefficients and standard errors.

Multiple linear regression analysis revealed that multiple rGMVs were significantly correlated with the subjective fatigue score of the CFQ. Figure 3 depicts the brain regions that were significantly correlated with CFQ scores. The positive and negative correlations in GMV based on the CFQ score were color-coded blue and red, respectively. Positive correlations mean the rGMV and the fatigue score got positive relations, i.e., the higher the rGMV, the higher the CFQ score. Conversely, a negative correlation means that the higher the rGMV, the lower the CFQ score.

## 4. Discussion

Fatigue studies have been conducted on severe and pathological cases, such as ME/CFS, but the number of participants with the disease was small. Long-term tiredness and fatigue due to the COVID-19 infection have recently emerged as major medical and social issues that must be addressed worldwide. Therefore, there has been a great deal of interest in research on fatigue severity or the early stages of fatigue progression. Nevertheless, fatigue studies using large-scale healthy adults were extremely few in the literature. Furthermore, only two studies used volumetric data with conventional MRI other than fMRI and PET studies. One study discovered that reducing the total GMV paralleled subjective fatigue scores in a healthy adult population [27]. However, the rGMVs were not examined, and the sample size (63 participants) was small. Another study by Tohoku University’s research team using young, healthy students (N = 883) discovered no correlation between rGMV and the degree of fatigue [6]. 

However, this present study identified two GM regions in all participants, three in the male, and two in the female, respectively, using 2955 healthy adults (Table 2). This difference may be due to the wide age range of the participants, who are more likely to experience fatigue at younger ages. Another reason may be that this study used a single MRI machine with the same protocol, which minimizes equipment bias.

We identified several regions that correlate with the CFQ scores and may be used to predict the CFQ scores (Table 2). These regions can explain the plausible fatigue relationship: the left caudate was positively correlated in all participants. In 150 patients with early-stage multiple sclerosis, the degree of fatigue was associated with a reduction in caudate volume [28]. This suggests that multiple sclerosis preferentially affects the caudate in some patients, resulting in fatigue in the early stages of the disease. 

In all participants and the male group, the right orbital part of the inferior frontal gyrus (OIFG) was negatively correlated. The orbitofrontal cortex extended into the laterally adjacent inferior frontal gyrus. There was abnormal functional connectivity between the inferior frontal gyrus and orbitofrontal cortex in 282 patients with major depressive disorder, including tiredness, and 254 controls [29]. This study suggests that the orbitofrontal cortex is involved in depression and influences mood and behavior via the inferior frontal gyrus. Furthermore, a recent study investigated the effect of age and sex in 43 healthy individuals on self-reported fatigue perception using the visual analog scale of fatigue (VAS-F) during a fatiguing task [30]. The results show that sex affected VAS-F under fatigue loading in several regions, including the OIFG.

The left precuneus and angular gyrus were negatively correlated in the male group. An fMRI study investigating 21 patients with bipolar depression and 20 healthy controls discovered that the left precuneus might be important for a favorable response to lamotrigine as a supplement for these patients [31]. Another fMRI study reported decreased connectivity between the left precuneus, angular gyrus, and right medial prefrontal cortex in patients with CFS [32]. A cross-sectional study of 87 non-hospitalized recovered individuals 54 days after laboratory confirmation of COVID-19 revealed that the individuals had symptoms of fatigue, anxiety, excessive somnolence, language impairment, and impaired cognitive flexibility [33]. Additionally, they discovered a severe impairment of the visuospatial network in the angular gyrus.

In the female group, the left middle temporal gyrus (MTG) was positively correlated. The decrease in MTG volume may be used as a candidate biomarker for schizophrenia. An fMRI study of 79 patients with multiple sclerosis (50 fatigued, 29 non-fatigued) and 26 matched healthy controls revealed that the left MTG was less activated compared with the controls [34]. In our study, the right inferior temporal gyrus (ITG) was negatively correlated with fatigue scores. In the first episode of schizophrenia, an MRI volumetric study revealed a smaller bilateral volume of the ITG [35], and an fMRI study revealed increased activation in fatigued patients with multiple sclerosis compared to healthy and non-fatigued patients [36].

Several GM regions positively interact with fatigue; that is, they expand in volume as one of the defenses against fatigue development because biological reactions often induce compensatory processes. Table 1 and Table 2 show that there are two rGMVs that have a negative correlation in males and one rGMV that has a negative correlation in females, with females having significantly higher fatigue scores and a higher TBV/ICV. This is an interesting result where although females reported higher fatigue scores compared to males, females also have a significantly higher TBV/ICV. These higher brain volumes in females might be the biological reactions of the brain as one of the defenses against fatigue development.

According to Watanabe [3], the fluctuation in some rGMVs with subjective fatigue suggests that the defense mechanisms in the pathophysiology of ME/CFS are activated to prevent further exhaustion. This statement is supported by small-scale studies of healthy individuals in a healthy state [37,38] and ME/CFS pathological state. The same hypothesis may be true for the positive correlation between the left caudate, angular gyrus, and MTG. Functional and metabolic imaging studies based on regional volumetric data from many healthy individuals are required to validate this hypothesis. 

This study demonstrated correlations between CFQ scores and regional gray matter volume. However, this study has several limitations, as follows: first of all, we did not exclude participants with comorbidities such as diabetes, hypertension, or/and dyslipidemia that were taking therapeutic drugs for these diseases because it is unclear whether chronic fatigue or ME/CFS is related to these diseases or drugs. In the near future, we will examine the relationship between common diseases, therapeutic drugs, and fatigue. The questionnaire scores lack biological parameters (such as stress level, anxiety, or sleep disorders) associated with fatigue development; the participants belong to a specific group of people from Kochi, Japan, which may have caused a selection bias; this study also only uses cross-sectional data, which cannot provide a causal effect of fatigue on rGMV.

Therefore, we plan to address these limitations in our future study by including accurate fatigue assessment and adding measurements of alpha-amylase activity in the salivary gland [39], autonomic nervous system activity [40], or inflammatory cytokines [41]. To validate the results of this study and investigate any causal effect of fatigue on rGMV, we plan to conduct longitudinal data analysis because we can then obtain MRI data over time during brain healthcare checkups in a less invasive manner. 

A volumetric study using regular MRI scans rather than fMRI and PET has a significant advantage for large-scale data collection in ordinary medical facilities where brain healthcare checkups are performed. Based on this longitudinal study using MRI volumetric data, the neuronal mechanisms underlying the early stages of fatigue and chronicity may be elucidated. If any evidence exists, the preclinical state of ME/CFS and long COVID-19 syndrome could be detected through regular MRI scans, and effective measures to reduce socioeconomic loss could be established. 

## Figures and Tables

**Figure 1 jcm-11-06037-f001:**
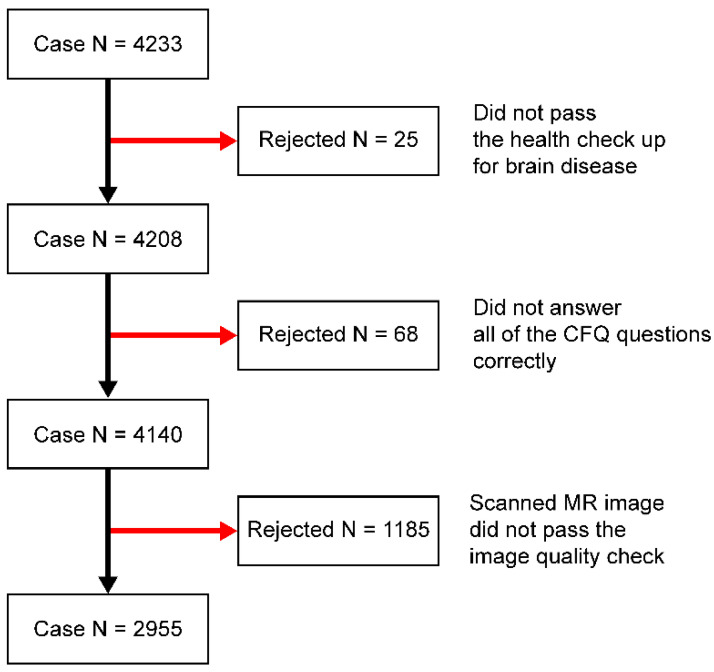
The participant-selection process.

**Figure 2 jcm-11-06037-f002:**
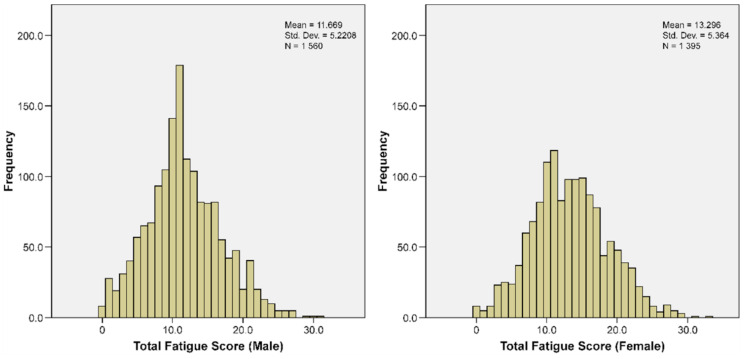
Distribution of the fatigue score for male participants (**left**) and female participants (**right**).

**Figure 3 jcm-11-06037-f003:**
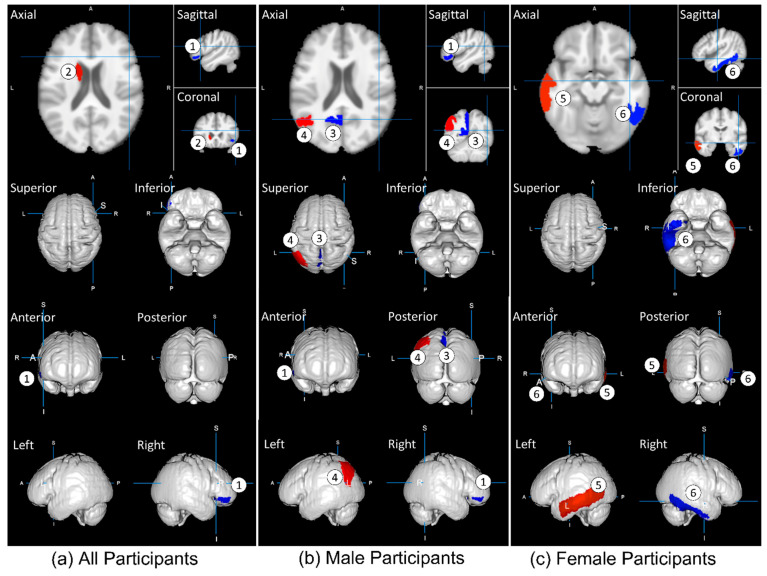
Impact of fatigue on the gray matter volume of the participants: (**a**) all participants; (**b**) male participants; (**c**) female participants. The most significant regions as predictors are presented in Table 2 and are color-coded with red for a positive correlation and blue for a negative one. The rGMVs are as follows: (1) right orbital part of the inferior frontal gyrus; (2) left caudate; (3) left precuneus; (4) left angular gyrus; (5) left middle temporal gyrus; (6) right temporal gyrus.

**Table 1 jcm-11-06037-t001:** Sex differences in age, Chalder’s fatigue questionnaire score, and several brain region volumes (divided and standardized based on intracranial volume).

	All Participants (N = 2955)		Male (N = 1560)		Female (N = 1395)		*p*-Value
	Mean	SD	Mean	SD	Mean	SD	
Age	53.1536	9.7437	53.2314	10.4482	53.0667	8.8929	>0.05
Chalder’s Fatigue Score	12.4372	5.3500	11.6692	5.2208	13.2961	5.3640	<0.001
Total Brain Volume/ICV	0.8233	0.0211	0.8182	0.0217	0.8296	0.0190	<0.001
Total GMV/ICV	0.4273	0.0209	0.4215	0.0213	0.4876	0.0185	<0.001
Total WMV/ICV	0.3960	0.0182	0.3968	0.0182	0.4521	0.0182	<0.05

Abbreviations: GMV, gray matter volume; WMV, white matter volume; ICV, intracranial volume; SD, standard deviation.

**Table 2 jcm-11-06037-t002:** Linear regression (stepwise-forward) analysis results with CFQ scores as the target variable and age and volume of the brain regions as the independent variables.

Parameters and Brain Regions that Significantly Predict CFQ Scores (Total Fatigue Score)		Coefficient (B)	Std. Error	*t*-Value	*p*-Value
All Participants (N = 2955)	(Constant)	17.554	1.344	13.066	<0.001
	Sex	−1.548	0.199	−7.796	<0.001
	Age	−0.074	0.011	−6.899	<0.001
	Right orbital part of the inferior frontal gyrus	−2.003	0.797	−2.513	<0.05
	Left caudate	1.073	0.436	2.460	<0.05
Male (N = 1560)	(Constant)	20.636	2.385	8.652	<0.001
	Age	−0.068	0.014	−4.733	<0.001
	Right orbital part of the inferior frontal gyrus	−3.006	1.080	−2.784	0.005
	Left precuneus	−0.808	0.254	−3.183	0.01
	Left angular gyrus	0.593	0.267	2.218	<0.05
Female (N = 1395)	(Constant)	16.195	2.606	6.214	<0.001
	Age	−0.081	0.017	−4.749	<0.001
	Left middle temporal gyrus	0.628	0.203	3.095	<0.01
	Right inferior temporal gyrus	−0.658	0.300	−2.194	<0.05

Abbreviation: Chalder’s Fatigue Questionnaire (CFQ); Standard Error (Std. Error).

## Data Availability

After publication, data will be available to any researcher who provides a methodologically sound study proposal to the corresponding author that is approved by the central study team. Individual participants will not be identifiable in any released data and all appropriate information governance.

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
