# Peer review of "Sex Differences in the Correlation between Fatigue Perception and Regional Gray Matter Volume in Healthy Adults: A Large-Scale Study"

_jcm, 2022, doi:10.3390/jcm11206037_

Round 1

Reviewer 1 Report

I read the manuscript ''Sex differences in the correlation between fatigue perception and regional gray matter volume in large-scale healthy adults'' carefully. The Paper is well written but I have only one comment, which could improve the Manuscript. 

* In the Abstract: In 16 females, the right inferior temporal gyrus was negative, whereas the left middle temporal gyrus and 17 right putamen were positive. It is unclear what is meant by ''positiv and negativ'' to the readers. 

Author Response

Dear Reviewer 1,

Thank you very much for your kind review.

In your review, you raise the following concern:  It is unclear what is meant by ''positive and negative'' to the readers. 

In order to make it clearer and easier to read, we revised the abstract (lines 14-20) and also Section 3.2. (lines 182-185)

Here we copied and pasted the revised abstract (the revised part is written in bold):

Abstract:

A higher CFQ score denotes a higher perceived fatigue level by the participant. According to the CFQ scores in males, the volumes of the right orbital part of the inferior frontal gyrus and left precuneus were negatively correlated (i.e., smaller rGMV for higher CFQ scores). In contrast, the left angular gyrus was positively correlated. In females, the right inferior temporal gyrus was negative, whereas the left middle temporal gyrus and right putamen were positive (i.e., larger rGMV for higher CFQ score).

Section 3.2 (lines 182-185):

The positive and negative correlations in GMV based on the CFQ score were color-coded blue and red, respectively. Positive correlations mean the rGMV and the fatigue score got positive relations, i.e., the higher the rGMV, the higher the CFQ score. Conversely, negative correlations mean that the higher the rGMV, the lower the CFQ score.

Reviewer 2 Report

Large data set, worth looking at. However, regarding the radiological parameters used for measurements, this paper may benefit from a reviewer from radiology background or functional MRI experts to cover that aspect in the review process.

information to be clarified:

1.       Did they exclude patients with small vessels disease (microangiopathy) on MRI?

2.       They need to discuss the available literature about the function of the area identified to correlate with fatigue (inferior frontal gyrus, angular gyrus, middle and inferior temporal).

3.       They need to explains the reasons why the area correlated in male are different than female? Are they different types of fatigue?

4.       Did they include or exclude patients with comorbidities such as Diabetes or hypertension or hypothyroidism or hyperthyroidism

5.       Were these participants taking any medications? Anti-hypertension, beta blockers, antidepressant?.

6.       Figure 3 needs better explanation in the caption or a better labeling on of the colored region in the figure.

Author Response

(we also provide our response in the attached document, thus please see the attachment)

Dear Reviewer,

Thank you very much for your kind review.

In your review, you raised several concerns. Please let us answer and revise our work according to your review.

Concern 1: Did they exclude patients with small vessel disease (microangiopathy) on MRI?

Response 1: In this study, we excluded small vessel disease, which includes small subcortical (lacunar) infarcts (of deep gray nuclei and deep white matter), hemorrhages, perivascular spaces, and diffuse white matter changes. We added the above sentence in the Methods (from 83 to 86 lines on page 2):

We excluded participants with small vessel disease, including small subcortical (lacunar) infarcts (of deep gray nuclei and deep white matter), hemorrhages, perivascular spaces, and diffuse white matter changes. We also exclude participants with hypothyroidism and hyperthyroidism and taking neurotropic drugs such as antidepressants.

Concern 2: They need to discuss the available literature about the function of the area identified to correlate with fatigue (inferior frontal gyrus, angular gyrus, middle and inferior temporal).

Response 2: all rGMVs that correlate with the fatigue score (CFQ score) are listed in Table 2. We have discussed the functions of the correlated rGMV with fatigue in Discussion section. You can find the discussion of the related rGMVs in the following lines of the manuscript:

  • Orbital part of inferior frontal gyrus: (lines 220-229), reference [28] and [29]: There was abnormal functional connectivity between the inferior frontal gyrus and orbitofrontal cortex in 282 patients with major depressive disorder, including tiredness, and 254 controls [28]. This study suggests that the orbitofrontal cortex is involved in depression and influences mood and behavior via the inferior frontal gyrus. Furthermore, a recent study investigated the effect of age and sex in 43 healthy individuals on self-reported fatigue perception using the visual analog scale of fatigue (VAS-F) during a fatiguing task [29]. The results show that sex affected VAS-F under fatigue loading in several regions, including the OIFG.
  • Left caudate: (lines 215-219), reference [27]:
    In 150 patients with the early stage of multiple sclerosis, the degree of fatigue was associated with a reduction in caudate volume [27]. This suggests that multiple sclerosis preferentially affects the caudate in some patients, resulting in fatigue in the early stages of the disease.
  • Left precuneus and left angular gyrus: (lines 230-235), reference [30][31][32]:
    An fMRI study investigating 21 patients with bipolar depression and 20 healthy controls discovered that the left precuneus might be important for a favorable response to lamotrigine as a supplement for these patients [30]. Another fMRI study reported decreased connectivity between the left precuneus, angular gyrus, and right medial prefrontal cortex in patients with CFS [31]. A cross-sectional study of 87 non-hospitalized recovered individuals 54 days after laboratory confirmation of COVID-19 revealed that the individuals had symptoms of fatigue, anxiety, excessive somnolence, language impairment, and impaired cognitive flexibility [32]. Additionally, they discovered a severe impairment of the visuospatial network in the angular gyrus.
  • Left and right middle temporal gyrus: (lines 241-247), reference [33][34][35]:
    An fMRI study of 79 patients with multiple sclerosis (50 fatigued, 29 non-fatigued) and 26 matched healthy controls revealed that the left MTG was less activated compared with controls [33]. In our study, the right inferior temporal gyrus (ITG) was negatively correlated with fatigue scores. In the first episode of schizophrenia, an MRI volumetric study revealed a smaller bilateral volume of the ITG [34], and an fMRI study revealed increased activation in fatigued patients with multiple sclerosis compared to healthy and non-fatigued patients [35].

Concern 3: They need to explain the reasons why the area correlated in male are different than female? Are they different types of fatigue?

Response 3: the main purpose of this study is to investigate if there is any difference between males and females in the correlation between regional brain volume and fatigue degree using the CFQ score, as mentioned in our Introduction section (lines 62-64):

this study aimed to analyze sex differences in the correlation between regional gray matter volume (rGMV) and fatigue degree using Chalder’s fatigue questionnaire (CFQ) scores on 2955 healthy adults without ME/CFS and COVID-19 infection.  

One of the past studies [reference 29] also suggests that there is indeed a different effect of fatigue on the human brain between males and females.

After our rigorous study and analyses, we confirmed that there are indeed differences between males and females in terms of correlation between regional brain volumes and fatigue scores. Although both males and females are subjected to similar levels of fatigue based on the CFQ score, the responses of the brains between males and females are different. In males, the CFQ scores, correlate significantly with the right orbital part of the inferior frontal gyrus, left precuneus, and left angular gyrus. While in females, it is the left and right middle temporal gyrus.

There are several plausible reasons why there is a difference in which brain regions correlate with fatigue between males and females, such as the difference in biological structure and hormonal effect. Further investigation is needed to understand why there are differences between males and females. However, this investigation is out of the scope of our manuscript. We have mentioned this limitation in our Discussion section (lines 260-268):

However, this study has several limitations as follows: we didn’t exclude the participants with comorbidities such as Diabetes, hypertension, or/and dyslipidemia and taking therapeutic drugs for these diseases because it is unclear whether chronic fatigue or ME/CFS is related to these diseases or drugs. In near future, we will examine the relationship between common diseases, therapeutic drugs, and fatigue. The questionnaire scores lack biological parameters (such as stress level, anxiety, or sleep disorders) associated with fatigue development; the participants belong to a specific group of people from Kochi, Japan, which may have caused a selection bias; and this study uses cross-sectional data only that cannot provide a causal effect of fatigue on rGMV.

Concern 4: Did they include or exclude patients with comorbidities such as diabetes or hypertension or hypothyroidism or hyperthyroidism?

Concern 5: Were these participants taking any medications? Anti-hypertension, beta-blockers, antidepressants?

Response to 4 and 5: We excluded participants with hypothyroidism and hyperthyroidism. We also excluded participants taking neurotropic drugs such as antidepressants. We added the above sentence in the Methods (from lines 85 to 86 on page 2):

We excluded participants with small vessel disease, including small subcortical (lacunar) infarcts (of deep gray nuclei and deep white matter), hemorrhages, perivascular spaces, and diffuse white matter changes. We also exclude participants with hypothyroidism and hyperthyroidism and taking neurotropic drugs such as antidepressants.

However, in the present study, we did not exclude participants with comorbid diseases such as diabetes, hypertension, or/and dyslipidemia. In the Discussion, we added the explanation as a limitation of the study as below (lines 260 to 264):

However, this study has several limitations as follows: we didn’t exclude the participants with comorbidities such as Diabetes, hypertension, or/and dyslipidemia and taking therapeutic drugs for these diseases because it is unclear whether chronic fatigue or ME/CFS is related to these diseases or drugs. In near future, we will examine the relationship between common diseases, therapeutic drugs, and fatigue.

Concern 6: Figure 3 needs better explanation in the caption or a better labeling on of the colored region in the figure.

Response to 6: Thank you for pointing out the readability of Figure 3. We will revise the Figure and its caption to make it easier for the reader to understand it. please find the revised figure in the attached document. (lines 189-194):

The caption of Figure 3: Impact of fatigue on the gray matter volume of the participants (a. all participants; b. male participants; c. female participants). The most significant regions are presented in Table 2 and are color-coded with red for a positive correlation and blue for a negative one. The rGMVs are as follows: 1. Right orbital part of the inferior frontal gyrus; 2. Left caudate; 3. Left precuneus; 4. Left angular gyrus; 5. Left middle temporal gyrus; 6. Right temporal gyrus.

Reviewer 3 Report

One of the large studies tried to understand the sex difference in fatigue and correlation with imaging findings using 1.5 teslas MRI. 

the study was done using CFQ questionnaire to assess the extent the fatiguability. 

The high celling effects of this scoring system lack the validity of this study.

Author Response

Dear Reviewer,

Thank you for your time and kind help to review our manuscript.

In the review, you raised a concern that the scoring system got a high ceiling effect.

However, we argue that Chalder's fatigue questionnaire (CFQ) score in our study does not have a ceiling effect. We presented the scoring distribution in Figure 2 for both males (left figure) and females (right figure). As can be seen from the figure, the CFQ score follows a normal distribution and does not have any ceiling nor flooring effect.

We hope our response answers your concern.

Best regards,

Authors

Round 2

Reviewer 2 Report

The authors have provided appropriate responses to my comments.

Author Response

Thank you very much for your kind review.